# Protein Quality and Sensory Perception of Hamburgers Based on Quinoa, Lupin and Corn

**DOI:** 10.3390/foods11213405

**Published:** 2022-10-28

**Authors:** Raquel Chilón-Llico, Lilia Siguas-Cruzado, Carmen R. Apaza-Humerez, Wilter C. Morales-García, Reynaldo J. Silva-Paz

**Affiliations:** 1Professional School of Human Nutrition, Faculty of Health Sciences, Universidad Peruana Unión, Lima 15472, Peru; 2Professional School of Food Industry Engineering, Faculty of Engineering, Universidad Peruana Unión, Lima 15472, Peru; 3Public Health Department, Postgraduate School, Universidad Peruana Unión, Lima 15472, Peru

**Keywords:** hamburger, vegetal nutrition, PDCAAS, sensory profile, amino acids

## Abstract

The need for partial or total substitution of animal protein sources by vegetable sources of high protein quality with good sensory acceptance is a promising alternative. The objective was to develop a hamburger with vegetable protein using a mixture design based on quinoa (*Chenopodium quinoa Willd.*), Peruvian Andean corn (*Zea mays*) and Andean lupine (*Lupinus mutabilis Sweet*). The design of these mixtures allowed obtaining eleven formulations, three of which were selected for complying with the amino acid intake for adults recommended by FAO. Then, a completely randomized design was applied to the selected samples plus a commercial product. Proximal composition was measured on a dry basis (protein, fat, carbohydrates, and ash), calculation of the Protein Digestibility Corrected Amino Acid Score (PDCAAS) and a sensory analysis was carried out using the Check-All-That-Apply (CATA) method with acceptability in 132 regular consumers of vegetarian products. Protein, fat, carbohydrate, and ash contents ranged from 18.5–24.5, 4.1–7.5, 65.4–72.1 and 2.8–5.9%, respectively. The use of Andean crops favored the protein content and the contribution of sulfur amino acids (SAA) and tryptophan from quinoa and lysine and threonine from lupin. The samples with Andean crops were described as easy to cut, soft, good, healthy, legume flavor, tasty and light brown, however the commercial sample was characterized as difficult to cut, hard, dark brown, uneven color, dry and grainy. The sample with 50% quinoa and 50% lupin was the most acceptable and reached a digestibility of 0.92. It complied with the lysine, threonine, and tryptophan intake, with the exception of SAA, according to the essential amino acid pattern proposed by the Food and Agriculture Organization of the United Nations.

## 1. Introduction

The United Nations indicates that plant-based diets are necessary to counteract climate change [1], which promotes the sustainability of natural resources [2]. This is due to the fact that 37% of cultivated land worldwide is used in food production to generate animal protein [3], generation of anthropogenic greenhouse gas (GHG) emissions [4] and loss of natural diversity [5]. In the last decades, interest in healthy plant-based nutrition has increased, since the consumption of red and processed meat is associated with the risk of suffering from various comorbidities, such as colon cancer, cardiovascular problems [6], chronic kidney disease [7], non-alcoholic fatty liver disease [8], chronic obstructive pulmonary disease (COPD), etc. [9]. However, despite the nutritional and sustainability benefits promised by plant-based diets, few people have adopted and adhered to this type of diet [10,11,12], in addition to indicating that the seemingly inappropriate taste is one of the main barriers to adopting this type of diet [13].

The hamburger is the most popular food in the culinary habits of the American diet. As such, there are health, environmental and moral value challenges of this fast food. In the face of these challenges, the growing trend in plant-based food production has provided companies to adapt to consumer demands and increase the offerings of these foods [14]. Technological innovations have been implemented to produce protein alternatives called meat substitutes or meat analogues [15,16], which allow different nutritional applications [17]. Although there are great challenges, from the choice that remains difficult for most consumers [18], the nutritional aspect that provides the reduction of saturated fats and guarantees protein quality [19], in addition to the sensory pleasure that seeks to improve the taste and texture of the final product [19]. This alternative is based on health care, ethical and moral issues [20].

The hamburger needs a strong protein structure, therefore the nutritional part focused on protein as an essential nutrient in the daily diet [17] that fulfills structural and functional roles in the human body [21,22]. Animal source protein is the most consumed and valued for its nutritional and protein quality, determined by its ability to meet the amino acid (AA) requirement [23], the level of digestion, absorption and its use for metabolic functions in the individual [24,25,26]. However, the use of mixtures of complementary protein sources in the production of modern analog products such as hamburgers could offer approximately the same nutrient composition as traditional meat products [27,28,29]. To know the protein quality there are several methods such as net protein utilization and the protein efficiency index, the protein digestibility corrected amino acid score (PDCAAS) method, is a practical way to ensure the adequacy of protein intake [28] regardless of whether the protein of each food is high or low. FAO/WHO recommends the use of the PDCAAS method to assess protein quality [30], which is calculated based on the estimation of protein digestibility and amino acid score based on a comparison with the standard (according to age).

Likewise, quinoa (*Chenopodium quinoa Willd.*), has a high nutritional value and contains 20 amino acids (including the 10 essential ones), so it has a high protein content (15.6–18.7%), is rich in phytosterols and n-3 and n-6 fatty acids, classifying it as the most complete cereal that allows it to compete with animal protein from meat, milk, and egg. In addition, it contains a low glycemic index, low level of fat, has no cholesterol and is suitable for coeliacs as it is gluten-free [31,32,33,34]. In the case of lupin (*Lupinus mutabilis*) is a legume that has a high protein content 46-48 (dm) and fat content 15.4% (dm) [35], is rich in dietary fiber and high in lysine [36], potassium, phosphorus, iron and tocopherols that remain high even after extrusion and atomization [37,38], making lupin an excellent alternative animal protein [39]. Peruvian Andean Cuzco Gigante maize (*Zea mays*) provides 8.94% protein that contributes with fibrous texture [40], digestible carbohydrates, 64.71, moisture 11.29%. Moreover, it is rich in SAA and magnesium, anthocyanin and carotenes, which benefit [41,42]. However, at present, there is little research on the proximal composition and sensory profile of plant-derived blends approaching protein quality.

On the other hand, consumers provide relevant information for product development, given that the inadequate taste of meatless diets are the main obstacle for the transition from meat foods to vegetarian alternatives [13], which reduces the purchase intention of plant-based and mixed (animal and vegetable protein) burgers, i.e., the introduction of mixed proteins could contribute towards the reduction of meat for those more attached [20,43]. Thus, meat analogues provide different gastronomic experiences, regarding cooking changes and losses are lower and the appearance of the prepared product, such as cutting effort is lower, color difference is detectable, etc. [44]. Despite the advancement of the food industry, meat-based burgers are more preferred than plant-based burgers [45], even compared to commercial samples [46], which is why further efforts are needed to boost the consumption of meat analogues.

In order to develop food products from Andean grain blends as ingredients to improve their sensory and nutritional properties, a specific formulation is required. Blend design is a methodology that has been applied to determine ideal proportions. Aguilar et al. [47] developed gluten-free breads based on quinoa, buckwheat and amaranth using the blending design. Paucar-Menacho et al. [48] reformulated breads using germinated pseudocereal grains to improve their nutritional value and sensory attributes. However, there are no reports on the study of a hamburger-type product from a mixture of Andean grains.

The objective of this study is to develop a hamburger-type product based on Andean crops (quinoa, lupin, and corn), traditionally consumed and easily accessible, in the form of amino-acid-complete food mixtures, evaluating their proximal composition, protein quality and sensory attributes.

## 2. Materials and Methods

### 2.1. Samples

Certified Quinoa (*Chenopodium quinoa Wild.*) grains, variety Blanca de Juli (Puno, Perú) and the certified maize variety INIA 618 Blanco Quispicanchi (Cusco, Perú) were obtained from the National Institute for Agrarian Innovation (INIA); the quinoa was de-bitterized with the technique of rubbing and manual washing [49] and corn was treated by peeling process with an alkaline substance (calcium oxide) (Lima, Peru), a method applied from pre-Inca times to the present day, as referred to by [50,51]. On the other hand, the Lupin (Puno, Peru) was obtained from the Khapía Wiñaymarka Association of agricultural producers in the town of Yunguyo and was debittered using the aqueous method of Jacobsen et al. [52], Erbas [53] and Córdova-Ramos et al. [54].

### 2.2. Formulation and Processing

The quinoa and corn were subjected to pressure cooking in an aqueous medium for 19 and 40 min, respectively. The lupin was ground manually in a Corona mill; then the three components were mixed, and in each formulation 20 g of base dressing was added per 100 g of mixture, which consisted of a blend of garlic, onion, tomato, paprika, vegetable oil and salt with 2 g, 3.5 g, 4 g, 4 g, 5 g and 1.5 g, respectively. The hamburgers were produced in a circular shape of 10 cm in diameter and 1 cm thick and sealed on the platten (baked), then cooled, packaged and frozen at −5 °C until analysis. The research consisted of three stages:

#### 2.2.1. Stage 1: Characterization of Inputs and Protein Quality

Proximal analysis of the main inputs: The physico-chemical analyses determined were moisture content (AOAC Method 925.10, 2000), protein by Kjeldahl method (AOAC 920.87, 2000), fat (AOAC Method 922.06, 2000), ash (AOAC Method 923.03, 2000) [55].

PDCAAS (Protein Digestibility Corrected Amino Acid Score): The amino acid contribution of the mixtures was measured, through the Amino Acid Score, based on the protein, amino acid and digestibility content of each main component (quinoa, lupin and corn) and finally the correction for the weighted average of the digestibility of the components was applied [30].

#### 2.2.2. Stage 2: Design of Hamburgers

Mixture design for Optimization: A simplex centroide mixture design was applied, with a total of 11 formulations (10 proposed by the program plus 1 added) and are shown in Table 1. Each component was denoted by x_1_, x_2_, …, x_q_, where x_i_ ≥ 0 for i = 1, 2, …, q and $\sum_{i=1}^{q}x_{i}=1=100\%$ and the function of the responses were identified as quadratic functions ($ŷ=\sum_{i=1}^{q}\beta_{i}x_{i}+\sum_{i\neq j}^{q}\beta_{\mathrm{ij}}x_{i}x_{j}$), the parameter β_i_ (main effects) of the function represents the expected response to the pure blend x_i_ = 100 and x_j_ = 0 when j ≠ i, β_ij_ the binary joint effects between the i^th^ and j^th^ components. The term $\;\sum_{i=1}^{q}\beta_{i}x_{i}$ represents the linear blending percentage. When curvature arises from non-linear blending between component pairs, the parameters β_ij_, which represent either antagonistic or synergistic blending, will be different from zero.

The contribution of the following amino acids was taken as a response variable: Lysine, Methionine and Cysteine (SAA), Tryptophan and Threonine. The influence of the components on the responses was observed through Cox traces.

#### 2.2.3. Stage 3: Proximal Analysis, Protein and Sensory Quality of Hamburger

From the 11 treatments obtained by optimizing the mixture design, three treatments were selected that meet the amino acid requirement according to FAO for adults (Lysine—45 mg/g, AAs—23 mg/g, Threonine—23 mg/g and Tryptophan—7 mg/g) (30), the selected treatments were subjected to the following analyses:

Proximal analysis: The physico-chemical analyses determined were moisture content (AOAC Method 925.10, 2000), protein by Kjeldahl method (AOAC 920.87, 2000), fat (AOAC Method 922.06, 2000), ash (AOAC Method 923.03, 2000) [55].

Sensory analysis for Check All That Apply (CATA): The 3 samples and a commercial sample (vegetarian hamburger based on white, red and black quinoa) were evaluated by 132 regular consumers of the ovo-lacto vegetarian canteen of a Private Clinic of Juliaca, before the evaluation the samples were heated on a gas griddle (150 °C) for 3 min on each side, the consumers received a primer consisting of acceptability (5-point hedonic scale) and 30 attributes, for the participants to indicate their level of liking and the attributes they considered necessary to describe the samples. The attributes were obtained from [44] with modifications.

Consumers signed a letter of consent for their participation. The study protocol was approved with the following code 2021-CEUPeU-0040 by the ethics committee of the Peruvian Union University and the vote 2021-025 Private Clinic.

Digestibility and aminogram: It was applied in the sample that presented greater acceptability, to determine the protein content the method according to Peruvian Technical Standard (NTP 205.005:2018) was used, regarding the aminogram the Analytical Biochemistry 136, 64.65 1984 methodology was applied, and the digestibility was determined by the method LMCTL-006F 2001 Analysis of Feed and Forages. MAX BECKER Then, the following formula was applied to obtain the percentage of digestibility of the sample: % Digestibility = [(Gr of residual protein without pepsin − Gr of residual protein with pepsin)/Gr of residual protein without pepsin] × 100.

### 2.3. Statistical Analysis

For the mixture design, the data obtained from the PDCAAS were used to perform an analysis of variance and a Cox plot. In the proximate composition results, the data were expressed as mean and standard deviation, the statistical assumptions were checked, and the analysis of variance was performed, and finally a multiple range test was applied. The sensory data obtained by the CATA method were subjected to Cochran’s test and correspondence analysis (CA). The data were processed using Design Expert version 11 and Xlstat Premium version 2022.

## 3. Results and Discussion

### 3.1. Formulation and Processing

#### 3.1.1. Stage 1: Proximate Chemical Composition of the Raw Material

Table 2 shows the values obtained from the proximate analysis of White Quinoa from Juli, Lupin and Corn variety INIA 618 White Quispicanchi.

The proximate composition showed that there were significant differences between the raw materials (*p* < 0.05). The debittered lupin (TD) showed higher moisture content (75.89%), although an inverse effect was observed in the debittered lupin (7.89%). In addition, the debittered lupin had higher fat and protein content. The highest carbohydrate content was found in corn, followed by quinoa with values of 85.08% and 73.03%, respectively. The moisture content was higher in the TD sample, this was, an expected value since the debittering process was performed in an aqueous medium [52,54] and the TSD sample had lower moisture content. As expected, the highest fat content was found in lupin, since it belongs to the legume family, and corn, a cereal, stood out for its higher carbohydrate content.

Regarding protein contribution, the TD showed significantly high values, higher than the TSD whose value coincides with [56] who analyzed different ecotypes of Andean lupin and found that the bitter seed of lupinus mutabilis contributes an average of 40.87% (32–46.9%) of protein, additionally the general correction factor used in the protein determination method is 6.25, and [57] stated that in case of legumes 5.5–5.7 should be used due to the high degree of amidation of legume proteins; the difference could be due to the method of unbittering. This difference coincides with the analyses of [58] who studied the effect of two debittering treatments and found that nitrogen content slightly increased or concentrated during hydration and the washing stage, which was attributed to the reduction of water-soluble compounds while protein remained inside the grain increasing its proportion. Regarding the protein content of quinoa is similar to those reported by [31] who analyzed Peruvian seeds harvested in southern Europe, finding a range of 15.6–18.7%. Moreover, [59] compared the proximate composition of Peruvian and Brazilian quinoa of different varieties finding a protein contribution between 12.5–16.9%, [60] analyzed a mixture of Kankolla and Blanca de Juli quinoa flour grown in 2017 in Puno (Peru), finding 13.5% protein. The protein content of maize was low as referred by [41,61] who also analyzed Peruvian Andean maize and found that the protein content varied from 7.28 to 9.64, with the Chullpi variety having the highest protein content and Cusco giant maize having the lowest content. Finally, there were no significant differences in ash content among the samples except for TSD.

Quinoa has a high protein content between 15.6–18.7% and fat 3.9–5.2 g·100 g^−1^ bs [31]. The seeds studied showed a high ash content (2.9–3.8 g·100 g^−1^ bs), and lower values of fructose and glucose, suggesting that the seeds have a low glycemic index [31]. Lupin (*Lupinus angosifolius*) flour was rich in protein (43 g/100 g) and dietary fiber (34 g/100 g) but low in carbohydrate (4.8 g/100 g) and ash (3.4 g/100 g) [36].

#### 3.1.2. Stage 2: Blend Design/Optimization and PDCAAS

References were taken from different authors, regarding protein content, amino acid contribution and digestibility factor, of the foods considered as raw material (quinoa, lupin and corn) that were used for PDCAAS calculation (Table 3): [62] studied quinoa, [41] analyzed corn from Cuzco and [63] evaluated lupin, registering values of 15.7, 7.1 and 55 g/100 g protein, lysine 48, 21.3 and 61.1 mg/g, SAA (methionine and cysteine) 33, 19 and 25.3 mg/g, threonine 32, 22.4 and 35.8 mg/g, tryptophan 11, 6.7 and 7.3 mg/g and digestibility 0.87, 0.82 and 0.9, respectively.

According Table 3, the range that the PDCAAS varied was between 0.71 and 1.59. Figure 1 shows traces that show the influence of the components on the response variables; the runs that meet the necessary score for adults were selected. It can be seen that the increase of quinoa and corn (to a lesser degree) favored the content of SAA, also the increase of lupin favored the content of lysine and threonine, on the other hand, the increase of quinoa favored the content of tryptophan. In addition, the effect of the addition of all components was of order 2 (quadratic).

The use of mixtures of plant-based foods contributes to the improvement of physical, chemical and sensory properties, although there are few studies on hamburgers based on mixtures of quinoa, lupin and corn, the results of this research reveal that the use of corn did not favor the contribution of any amino acid, which is similar to that reported by [64] who evaluated the amino acid profile and digestibility of ready-to-eat breakfast cereals based on rice, corn and oats, it was found that rice and corn favored digestibility but SAA were limiting for 50% of the cereals, followed by lysine (33%) and tryptophan (16%). The use of Andean crops favored the protein content and the contribution of SAA and tryptophan by quinoa and the contribution of lysine and threonine by lupin, in this regard [65] developed gluten-free breads, based on Andean grains and found that the optimized formulations were quinoa (46.3%), kiwicha (40.6%), kañiwa (100%) and lupin (12%) flours; on the other hand an optimized formulation of cookies with a good source of essential amino acids, containing 30% quinoa flour, 25% quinoa flakes and 45% corn starch [66]. Cereal and legume mixtures contribute with nutritional value such as protein and amino acid content. Such is the case of [67] who elaborated cakes based on mixtures of rice semolina flour, corn starch and extruded bean flour to replace wheat flour; they obtained products of high nutritional value, high protein content without gluten, without limiting the amino acid content. On the other hand, there are studies on the use of concentrated protein, mixtures of concentrated milk protein (PCL) and isolated soy protein, and rice. The lowest digestibility-corrected amino acid score was associated with rice protein. Furthermore, vegetable protein mixtures with PCL had higher digestibility in vitro [68]. In Nigeria it was possible to improve the nutritional qualities of feed mixtures such as protein and some amino acids such as leucine which was the most abundant amino acid and tryptophan the lowest content [69]. It was possible to improve the nutritional value and even the flavor with higher amounts of oats that favored sensory fibrosity, an additional highlight that legume flavor predominated [70].

It is important to highlight the use of foods available in a region. In this regard the content of nutrients such as protein (maximize) in a complementary food based on foods available in Nigeria was optimized and reached 20.26% protein and 377.21 Kcal of energy value. Complementary food products formulated from cereals, legumes and animal proteins have the potential to meet the macronutritional needs of infants and young children [71]. In Ethiopia, a formulation from 45% malted sorghum, 26% bleached soybean and 19% boiled karkade seed meal plus 10% premix was found to have better nutrient profiles and, in a presentation as a porridge is sensorially acceptable, to reduce malnutrition; furthermore at higher proportions of bleached soybean and karkade seed meal, protein, fat, energy and mineral contents were increased [72]. Mixtures of sorghum, African yam and soybean meal in the formulation of low-cost nutritious supplemental diets had an increase in nutrients, higher amounts of African yam and soybean had more crude protein [73].

#### 3.1.3. Stage 3: Proximal Analysis of Selected Hamburgers

Table 4 describes the values found for the proximal composition of the selected hamburgers. The moisture content for the samples was similar; it should be noted that the samples containing more quinoa and corn reported higher moisture content, which could be attributed to the carbohydrate (starch) content they provide. The protein and fat content was higher in the sample containing more lupin, this result was expected; the carbohydrate content is higher in the presence of corn and quinoa. The ash content was higher in the sample containing more quinoa. It was found that the protein content of the most acceptable hamburger (M05) was 24.60% (BS), which exceeds the values reached by [74,75,76] who found a lower value of 11.6% and 18.01% (respectively) in commercial plant-based hamburgers.

On the other hand, the fat content was lower (7.53%) than the values obtained by [74] who found total fat values of 8.58% and 11.10% respectively. This is due to the composition of the product itself, since it is made based on cereals and legumes that are low in fat, and mean a healthier alternative [75,77] mention that legumes can bind fat and water producing a firm texture after thermal processing, acting as binders, fillers and also improvers in the elaboration of hamburgers or other meat analogues due to their great versatility. Several commercial products of different styles can be found in the market, home-made, flour-based, concentrated and isolated protein (similar to meat), among others; most of the protein-based products are considered as a source of protein due to the nature of their ingredients and processing (mixtures, extrusion, cooking, etc.) that could improve digestibility and destroy some amino acids such as lysine. [18] found that home-made products or products made with fresh or minimally processed ingredients (legumes and vegetables) have a minimal contribution of protein (5%) but a significant contribution of carbohydrates and fiber (median of 23.5% and 4.4% respectively) unlike products made with isolated protein that stand out for being a source of protein (13%) and lower carbohydrate content (9.67%) in the study it was also found that for all cases the fat content was lower compared to meat (2.66–9.23%); in addition they suggest that the protein content of meat substitutes should be equivalent to the protein content of the legume that originated them, and not similar to that of meat. The most commonly used ingredients in these commercial products are soy protein, gluten and legumes (whole grains and flours), vegetable oil (soybean, sunflower, olive, coconut and palm) as a source of fat, and wheat flour as a source of carbohydrates that provide consistency and texture.

### 3.2. TASTING/Acceptability

The group of participants consisted of 132 consumers (56.1% male and 43.9% female) from the city of Juliaca in the Altiplano region of Peru at 3825 m.a.s.l. and 65% RH%. The nature and objective of the study was explained to each participant, and informed consent was requested. The majority of the participants were in the age range of 25–44 years (56%) and 87.9% of the consumers habitually consume vegetarian products.

A commercial quinoa-based sample was used for comparison with the optimized samples. Figure 2 shows the results of acceptability of the samples, which were rated as I like them to, I like them very much (scale 4 and 5). The samples showed significant differences using the LSD Fisher mean comparison test, where samples M05 and M11 achieved the highest acceptability. This can be attributed to the higher lupin content in the hamburger, which confers a more bitter and astringent flavor, characteristics of lupin [78]. In addition, consumers are familiar with the taste of quinoa, and M05 was selected for amino acid and digestibility analysis because it is more practical to use equal proportions for its preparation in the culinary field, and it could encourage the consumption of locally produced lupine, Adeniyi Paulina & A., (2018) studied the sensory properties of beef, chicken and soy burgers finding that all samples were acceptable, although they prefer chicken burgers. On the other hand, [46] studied beef, vegetable and hybrid burgers consumers’ sensory evaluation of three burgers: 100% beef, 100% vegetable and a hybrid (60% beef and 40% vegetable) where consumers showed in terms of acceptability, purchase intention and subjective comments positive results (I like it and I would buy it).

Figure 3 presents the correspondence analysis obtained by applying the CATA method, which explains 97.02% of the total variability of the data. Consumers formed three groups with the samples evaluated. The first group consisted of sample 343 described as hard, difficult to cut, dry and uneven color. The second group 459 was perceived as processed, greasy, seasoning, and salty. The third group samples 284 and 625 were characterized as easy to cut, soft, good, healthy, legume flavor, tasty, light brown color. Similar behavior was observed in ideal hamburger based on meat, vegetables and condiments described as tender or soft, natural, delicious, nutritious, healthy and tasty, in addition [79]. In addition, [80] optimized hamburger formulations based on mechanically separated tambaqui (*Colossoma macropomum*) meat with oat flour and cassava starch, finding that all samples were acceptable (I like it very much to I like it very much). However, the binary samples were rated superior in the preference test.

Table 5 shows the results obtained by Cochran’s Q test. Of the 30 descriptors used, 16 were significant, i.e., they showed significant differences. The commercial sample showed higher frequency in the terms dry, hard, bland, difficult to cut, dark brown color, grainy and uneven color; and lower frequency in light brown color, easy to cut, legume flavor, unlike the other samples, which were similar to each other for these terms. Sample 459 showed lower frequency in cereal flavor and higher frequency in salty, seasoning and juicy. This can be attributed to the fact that the inputs in hamburgers (fat substitution or addition of ingredients such as mushrooms) affect the texture, resulting in less tough and chewy products, but allow to characterize the hamburgers sensorially as juicy, tender and tasty. In addition, the hedonic tests indicated a good acceptance of this type of product [81].

### 3.3. Digestibility, Protein and Aminogram

Digestibility and amino acid content were evaluated in the sample that was most acceptable (50% quinoa and 50% lupin), the digestibility was good (0.91) and according to Table 6, the contribution of amino acids was acceptable except for Valine and SAA, for which the cysteine contribution could not be measured. In this respect, it is inferred that it is due to the treatment applied in the preparation (baking at 150 °C), on the other hand, this need could be supplied since a hamburger-type product is generally accompanied by cereals such as rice or wheat-based bread, which are a source of SAA.

## 4. Conclusions

Incorporating plant-based foods into culinary habits would provide important nutritional benefits. Hamburgers made from Andean grains such as quinoa, lupin and corn, substituting meat, can provide a source of complementary protein and provide high nutritional value. However, there are few studies on vegetarian mixtures and protein quality. Therefore, a better evaluation of Andean grains to form food masses will allow an optimized formulation of a hamburger-type product.

In addition, the results obtained in this study provide valuable data for the subsequent development of a textured vegetable protein as a vegan substitute for hamburger meat based on quinoa, corn, and lupin. Likewise, texturing a meat analog that is close enough to that used in hamburgers requires more research.

This first study is an approach to developing a textured vegetable meat. The hamburger presented differences in its protein content and amino acid composition through the application of the mixture design.

The selected hamburgers have an adequate proximal composition and digestibility. Processing (baking) interferes especially in the contribution of SAA but improves digestibility. The sensory attributes most used to describe the hamburgers were: easy to cut, soft, good, healthy, legume flavor, tasty and light brown color. It should be considered that there are several ways of preparing hamburgers, use of additives, ingredients or cooking methods that can be studied to elaborate vegetarian hamburgers based on products that are easily accessible and that are close to the amino acid profile recommended for adults.

## Figures and Tables

**Figure 1 foods-11-03405-f001:**
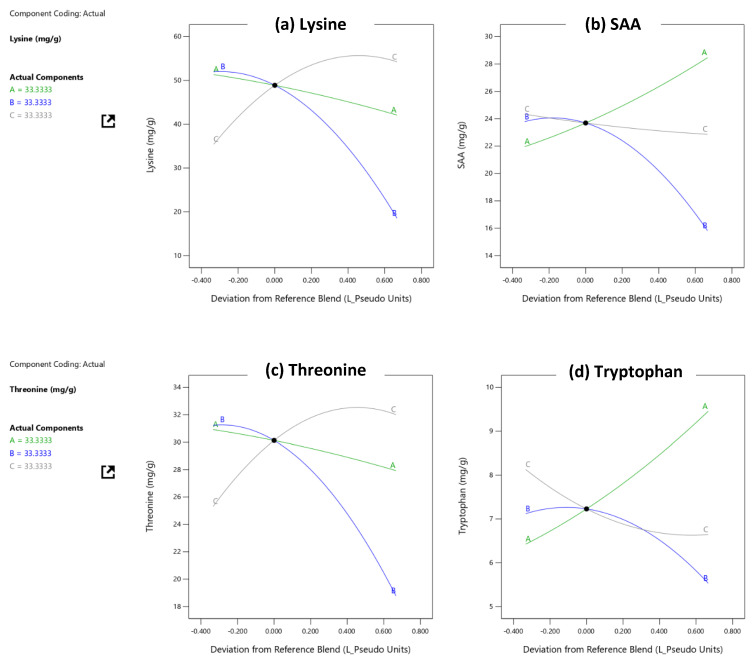
Influence of the raw material (Quinoa, Corn and Lupine) about supply of amino acids, according to the list: (**a**) Lysine; (**b**) Sulfur amino acids (SAA); (**c**) threonine and (**d**) tryptophanentonce.

**Figure 2 foods-11-03405-f002:**
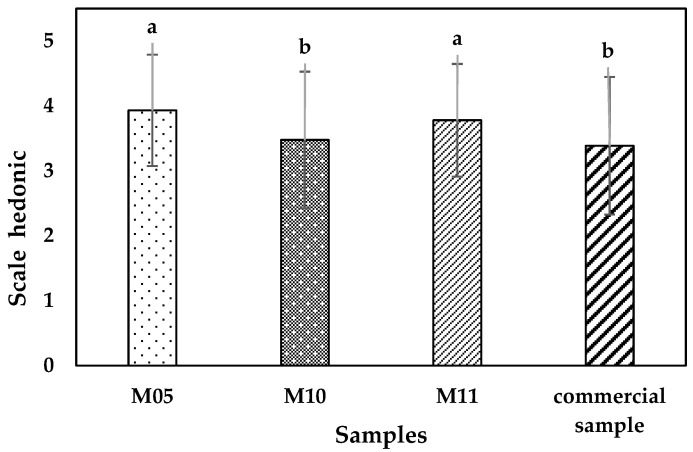
Comparison of means with the LSD Fisher Test (*p* value < 0.05). a, b: means of the samples.

**Figure 3 foods-11-03405-f003:**
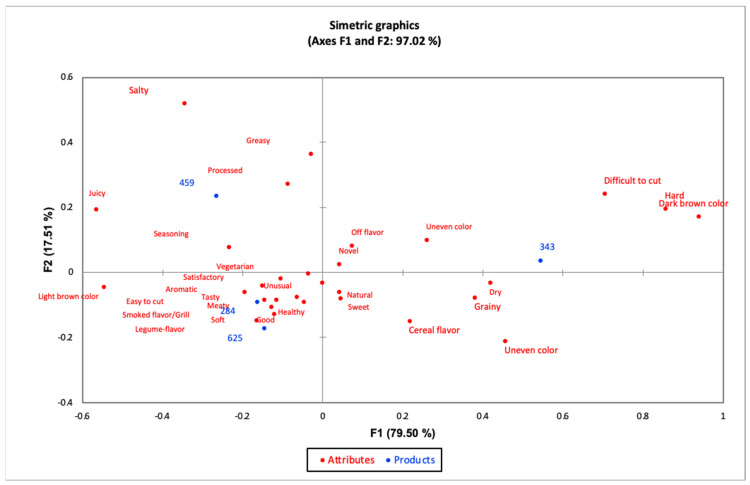
Correspondence analysis graph of hamburgers. 459 (33.33% quinoa, corn and lupine), 343 (commercial sample based on quinoa), 284 (90% quinoa and 10% lupine) and 625 (50% quinoa and 50% lupine).

**Table 1 foods-11-03405-t001:** Augmented simplex mix design.

Treatments	Quinoa (x_1_)	Corn (x_2_)	Lupin (x_3_)
	%		
M01	100.00	0.00	00.00
M02	00.00	100.00	00.00
M03	00.00	00.00	100.00
M04	50.00	50.00	00.00
M05	50.00	00.00	50.00
M06	00.00	50.00	50.00
M07	66.67	16.67	16.67
M08	16.67	66.67	16.67
M09	16.67	16.67	66.67
M10	33.33	33.33	33.33
M11	90.00	0.00	10.00

**Table 2 foods-11-03405-t002:** Proximal composition of raw material.

Proximal on Dry Basis	Quinoa (Q)	Debittered Lupine (TD)	Corn (M)	Lupine without Debittering (TSD)
	Mean ± SD (%)	Mean ± SD (%)	Mean ± SD (%)	Mean ± SD (%)
Moisture	17.22 ± 0.071 ^b^	75.890 ± 0.057 ^a^	12.910 ± 0.042 ^c^	7.895 ± 0.007 ^d^
Fat *	6.765 ± 0.006 ^c^	18.291 ± 0.016 ^a^	4.581 ± 0.018 ^d^	16.611 ± 0.001 ^b^
Protein *	17.637 ± 0.053 ^c^	61.448 ± 0.174 ^a^	8.152 ± 0.004 ^d^	43.602 ± 0.003 ^b^
CHOs *	73.037 ± 0.023 ^b^	17.959 ± 0.101 ^d^	85.084 ± 0.041 ^a^	34.922 ± 0.026 ^c^
Ash *	2.507 ± 0.006 ^b^	2.302 ± 0.259 ^b^	2.205 ± 0.064 ^b^	4.864 ± 0.03 ^a^

SD = Standard deviation; * = Wet basis; Different letters in the same row represent significant differences (*p* value < 0.05).

**Table 3 foods-11-03405-t003:** Mix design and PDCAAS.

Treatments	Lysine	SAA	Threonine	Tryptophan	Lysine	SAA	Threonine	Tryptophan
	mg/g				PDCAAS (Adults)			
M01	41.70	28.67	27.80	9.56	0.93	1.30	1.21	1.59
M02	17.47	15.58	18.37	5.49	0.39	0.71	0.80	0.92
M03	55.22	22.87	32.36	6.60	1.23	1.04	1.41	1.10
M04	33.88	24.45	24.76	8.25	0.75	1.11	1.08	1.37
M05	52.14	24.20	31.32	7.28	1.16	1.10	1.36	1.21
M06	50.57	21.98	30.64	6.47	1.12	0.90	1.33	1.08
M07	46.06	25.40	29.20	8.04	1.02	1.15	1.27	1.34
M08	41.63	21.61	27.41	6.74	0.93	0.98	1.19	1.12
M09	53.13	23.03	31.61	6.76	1.18	1.05	1.37	1.13
M10	48.74	23.36	30.06	7.11	1.08	1.06	1.31	1.18
M11	45.40	27.10	29.05	8.75	1.01	1.23	1.26	1.46

SAA = Sulfured Amino acids. PDCAAS = Protein digestibility-corrected amino acid score, M = Sample

**Table 4 foods-11-03405-t004:** Proximal composition of hamburger samples.

Sample	M05	M10	M11	Commercial Sample
Moisture	19.69 ± 0.19 ^c^	64.58 ± 0.07 ^b^	66.01 ± 0.03 ^a^	65.37 ^b^
Protein (db)	24.60 ± 1.3 ^a^	19.83 ± 0.05 ^b^	18.51 ± 0.04 ^b^	10.42 ^c^
Fat (db)	7.53 ± 0.07 ^a^	5.05 ± 0.06 ^b^	4.12 ± 0.05 ^c^	2.17 ^d^
Carbohydrates (db)	65.45 ± 1.15 ^c^	72.13 ± 0.2 ^b^	71.49 ± 0.01 ^b^	85.13 ^a^
Ash (db)	2.81 ± 0.17 ^b^	3.01 ± 0.17 ^b^	5.88 ± 0 ^a^	2.26 ^c^

db, dry basis. Superscript with different letters in the same row represent significant differences (*p* value < 0.05).

**Table 5 foods-11-03405-t005:** Frequency of mention of CATA attributes.

Attributes	*p*-Value	M11-284 (90% Quinoa and 10% Lupine)	Commercial Sample 343	M10-459 (33.33% Quinoa, Corn and Lupine)	M05-625 (50% Quinoa and 50% Lupine)
Dark brown color	0.000	0.114 ^(c)^	0.765 ^(a)^	0.136 ^(b)^	0.114 ^(c)^
Light brown color	0.000	0.538 ^(a)^	0.030 ^(b)^	0.523 ^(a)^	0.492 ^(a)^
Uneven color	0.021	0.106 ^(b)^	0.212 ^(a)^	0.129 ^(b)^	0.114 ^(b)^
Smoked flavor/Grill	0.274	0.182 ^(a)^	0.121 ^(a)^	0.152 ^(a)^	0.182 ^(a)^
Aromatic	0.204	0.220 ^(a)^	0.167 ^(a)^	0.227 ^(a)^	0.258 ^(a)^
Difficult to cut	0.000	0.076 ^(b)^	0.303 ^(a)^	0.098 ^(b)^	0.045 ^(b)^
Easy to cut	0.000	0.659 ^(a)^	0.409 ^(b)^	0.621 ^(a)^	0.705 ^(a)^
Legume Flavor	0.016	0.167 ^(ab)^	0.129 ^(b)^	0.167 ^(ab)^	0.258 ^(a)^
Cereal flavor	0.000	0.326 ^(a)^	0.402 ^(a)^	0.159 ^(b)^	0.280 ^(ab)^
Sweet	0.451	0.061 ^(a)^	0.045 ^(a)^	0.030 ^(a)^	0.030 ^(a)^
Salty	0.000	0.280 ^(b)^	0.212 ^(b)^	0.765 ^(a)^	0.242 ^(b)^
Bland	0.000	0.091 ^(b)^	0.212 ^(a)^	0.038 ^(b)^	0.136 ^(ab)^
Seasoning	0.000	0.371 ^(ab)^	0.235 ^(b)^	0.462 ^(a)^	0.379 ^(ab)^
Meaty	0.245	0.159 ^(a)^	0.167 ^(a)^	0.174 ^(a)^	0.235 ^(a)^
Tasty	0.079	0.402 ^(a)^	0.303 ^(a)^	0.348 ^(a)^	0.432 ^(a)^
Off flavor	0.921	0.045 ^(a)^	0.061 ^(a)^	0.053 ^(a)^	0.045 ^(a)^
Dry	0.000	0.242 ^(a)^	0.629 ^(b)^	0.220 ^(a)^	0.341 ^(a)^
Hard	0.000	0.053 ^(a)^	0.273 ^(b)^	0.061 ^(a)^	0.038 ^(a)^
Juicy	0.000	0.220 ^(ab)^	0.015 ^(c)^	0.250 ^(a)^	0.121 ^(b)^
Soft	0.105	0.152 ^(a)^	0.121 ^(a)^	0.144 ^(a)^	0.197 ^(a)^
Grainy	0.000	0.318 ^(b)^	0.553 ^(a)^	0.174 ^(c)^	0.273 ^(bc)^
Greasy	0.308	0.038 ^(a)^	0.053 ^(a)^	0.076 ^(a)^	0.030 ^(a)^
Good	0.002	0.432 ^(ab)^	0.326 ^(b)^	0.364 ^(ab)^	0.515 ^(a)^
Processed	0.230	0.076 ^(a)^	0.076 ^(a)^	0.114 ^(a)^	0.053 ^(a)^
Healthy	0.030	0.538 ^(a)^	0.477 ^(a)^	0.447 ^(b)^	0.583 ^(a)^
Satisfactory	0.528	0.235 ^(a)^	0.205 ^(a)^	0.250 ^(a)^	0.273 ^(a)^
Natural	0.083	0.462 ^(a)^	0.508 ^(a)^	0.394 ^(a)^	0.492 ^(a)^
Unusual	0.888	0.295 ^(a)^	0.295 ^(a)^	0.265 ^(a)^	0.295 ^(a)^
Vegetarian	0.910	0.515 ^(a)^	0.485 ^(a)^	0.492 ^(a)^	0.500 ^(a)^
Novel	0.320	0.402 ^(a)^	0.439 ^(a)^	0.379 ^(a)^	0.356 ^(a)^

Different letters in the same row indicate a significant difference according to the Bonferroni test.

**Table 6 foods-11-03405-t006:** Comparison of the amino acid intake of hamburger (M05) with the proposed FAO requirement.

Amino Acid	mg AA/g Protein	Req FAO
Threonine	58.06	23
Valina	30.65	39
Methionine	6.45	22
Isoleucine	77.74	30
Leucine	127.10	54
Phenylalanine + Tyrosine	50.00	38
Lysine	51.61	45
Tryptophan	6.45	6

Cysteine content is not reported.

## Data Availability

Data are available upon request to the corresponding author.

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
