# Peer review of "Protein Quality and Sensory Perception of Hamburgers Based on Quinoa, Lupin and Corn"

_foods, 2022, doi:10.3390/foods11213405_

Round 1

Reviewer 1 Report

This article "Protein quality and sensory perception of hamburgers based on quinoa (Chenopodium quinoa), lupin (Lupinus mutabilis) and corn (Zea mays)” was revised and has a novelty and I recommend it for publication after consideration of the following comments.

Title: If you can rewrite and make it more interesting for readers. I propose: “Protein quality and sensory perception of hamburgers based on quinoa, lupin and corn”.

Abstract:

·       The type of statistical design used in this research should be mentioned.

·       Explain the statistical design of mixer design in detail.

·       Line 24: Abbreviation of sulfur amino acids (AAS) or (SAA)?

Keywords: Please choose keywords other than the main words of the title. In this case, other researchers can find your article by searching a wide range of words through databases. I propose another keywords as the follow:

hamburger; vegetarian; PDCAAS; amino acids; sensory profile

Abbreviation:

·       Please provide “Abbreviation section consequent the Keywords

Introduction:

·        In the last paragraph of the introduction, briefly state the statistical design used and also the treatments used in the research

·       Line 82-83:The standard error or standard deviation should not be used in the entire text of the results

Materials:

·                   Line 102: Please write materials as Company Name (City, Country), especially for chemical analysis assessment which used in the study.

Methodology:

·       Line 132-136: The statistical design of Design Expert should be mentioned well and in detail.

·       Line 124-125 etc., the way of expressing the method of measuring macronutrients and other parameters has a scientific flaw. Please take help from the following article for the correct way of expressing it, so that the standard number of the working method should be clearly stated.

·       Line 142-144: please consider these sentences in results and discussion section.

·       2.3. Statistical analysis: please explain as detail of your design.

 Results:

·       Table 3 should be statistically compared in each column

·       Line 234-235: please change to English approach.

·       Table 4: Statistical comparison in each row is incomplete

·       All Tables especially Table 5: The alphabetical statistical letters for the means should all be modified such that the greatest number has the letter a and as the numbers go lower, letters b, c etc.

Discussion:

·       Discussion text must grammar improve and in some cases it is very weak and maybe there is no discussion at all.

·       Fig 2 : It seems that there should not be a significant statistical difference between M05 and M10 as well as between M11 and commercial sample. Please check the statistics again

Conclusions:

·       Conclusion is very general, try to make it more scientific, comprehensive and concise in detail, especially.

References:

·       The references in the reference list should be corrected according to the journal's instructions.

The article has many flaws in express and concept of English, it is suggested to be revised in a scientific and native way.

Author Response

Response to Reviewer 1 Comments

This article "Protein quality and sensory perception of hamburgers based on quinoa (Chenopodium quinoa), lupin (Lupinus mutabilis) and corn (Zea mays)” was revised and has a novelty and I recommend it for publication after consideration of the following comments.

Title: If you can rewrite and make it more interesting for readers. I propose: “Protein quality and sensory perception of hamburgers based on quinoa, lupin and corn”.

  1. Thank you for your comment, the suggested changes have been made.

Abstract:

  • The type of statistical design used in this research should be mentioned.
  1. Thank you for your suggestion the indicated changes were made
  • Explain the statistical design of mixer design in detail.
  1. Thanks for your suggestion has also been added
  • Line 24: Abbreviation of sulfur amino acids (AAS) or (SAA)?
  1. Thanks for your suggestion the changes were made

Keywords: Please choose keywords other than the main words of the title. In this case, other researchers can find your article by searching a wide range of words through databases. I propose another keywords as the follow:

hamburger; vegetarian; PDCAAS; amino acids; sensory profile

  1. Thanks for your suggestion the changes were made

Abbreviation:

  • Please provide “Abbreviation section consequent the Keywords

Thanks for the comments, they were added at the end of the document:

  1. PDCAAS: Protein Digestibility Corrected Amino Acid Score
  2. AA: Amino acids
  3. AAS: Sulfured Amino acids
  4. AOAC: Association of Analytical Communities
  5. CATA: Check-All-That-Apply
  6. FAO: Food and Agriculture Organization of the United Nations
  7. WHO: World Health Organization
  8. GHG: Emissions of greenhouse gases
  9. COPD: Chronic obstructive pulmonary disease
  10. INIA: National Institute for Agrarian Innovation
  11. Q: Quinoa
  12. T: Lupin
  13. TD: Debittered lupine (TD)
  14. M: Corn
  15. TSD: Lupine without debittering
  16. PCL: Concentrated milk protein
  17. M01: Sample 01

Introduction:

  • In the last paragraph of the introduction, briefly state the statistical design used and also the treatments used in the research
  1. Thanks for the comments, the following paragraph was added

“In order to develop food products from Andean grain blends as ingredients to improve their sensory and nutritional properties, a specific formulation is required. Blend design is a methodology that has been applied to determine ideal proportions. Aguilar et al (47) developed gluten-free breads based on quinoa, buckwheat and amaranth using the blending design. Paucar-Menacho et al (48) reformulated breads using germinated pseudocereal grains to improve their nutritional value and sensory attributes. However, there are no reports on the study of a hamburger-type product from a mixture of Andean grains.”

  • Line 82-83:The standard error or standard deviation should not be used in the entire text of the results
  1. Thanks for the feedback, corrections have been made.

Materials:

  • Line 102: Please write materials as Company Name (City, Country), especially for chemical analysis assessment which used in the study.
  1. Thanks for the feedback, corrections have been made.

Methodology:

  • Line 132-136: The statistical design of Design Expert should be mentioned well and in detail.
  1. Thanks for the comments, the indicated was added
  • Line 124-125 etc., the way of expressing the method of measuring macronutrients and other parameters has a scientific flaw. Please take help from the following article for the correct way of expressing it, so that the standard number of the working method should be clearly stated.
  1. Thanks for the feedback, We couldn't find the right suggestion, however the changes were made
  • Line 142-144: please consider these sentences in results and discussion section.
  1. Thanks for the feedback, your suggestion was considered. The part in parentheses refers to the amino acid requirement for adults according to the FAO (not the results). In addition, the text was adjusted so that it is consistent with the following paragraph.

“From the 11 treatments obtained by optimizing the mixture design, three treatments were selected that meet the amino acid requirement according to FAO for adults (Lysine - 45 mg/g, AAs - 23 mg/g, Threonine 23 mg/g and Tryptophan - 7 mg/g) (30), the selected treatments were subjected to the following analyses:

  • 2.3. Statistical analysis: please explain as detail of your design.

  1. Thank you for your comment, the following paragraph was added

“For the mixture design, the data obtained from the PDCAAS were used to perform an analysis of variance and a Cox plot. In the proximate composition results, the data were expressed as mean and standard deviation, the statistical assumptions were checked and the analysis of variance was performed, and finally a multiple range test was applied. The sensory data obtained by the CATA method were subjected to Cochran's test and correspondence analysis (CA). The data were processed using Design Expert version 11 and Xlstat Premium version 2022.”

 “Results:

  • Table 3 should be statistically compared in each column
  1. Thank you for your comments, however, the comparison was not made, it does not contain repetitions because it is a result of the PDCAAS calculation.
  • Line 234-235: please change to English approach.
  1. Changes were made
  • Table 4: Statistical comparison in each row is incomplete
  1. Thank you for your comment, Changes were made to the document
  • All Tables especially Table 5: The alphabetical statistical letters for the means should all be modified such that the greatest number has the letter aand as the numbers go lower, letters b, c etc.
  1. Thank you for your comments Changes were made to table 5

Discussion:

  • Discussion text must grammar improve and in some cases it is very weak and maybe there is no discussion at all.
  1. Thanks for your comments the changes were made·       

Fig 2 : It seems that there should not be a significant statistical difference between M05 and M10 as well as between M11 and commercial sample. Please check the statistics again

  1. Thanks for your comments changes were made in the following paragraphs

“The samples showed significant differences using the LSD Fisher mean comparison test, where samples M05 and M11 achieved the highest acceptability. This can be attributed to the higher lupin content in the hamburger, which confers a more bitter and astringent flavor, characteristics of lupin (78) In addition, consumers are familiar with the taste of quinoa, and M05 was selected for amino acid and digestibility analysis because it is more practical to use equal proportions for its preparation in the culinary field, and it could encourage the consumption of locally produced lupine, Adeniyi Paulina & A., (2018) studied the sensory properties of beef, chicken and soy burgers finding that all samples were acceptable, although they prefer chicken burgers.”

Conclusions:

  • Conclusion is very general, try to make it more scientific, comprehensive and concise in detail, especially.
  1. Thanks for your comments, the suggested changes were made.
  2. References:
  • The references in the reference list should be corrected according to the journal's instructions.

Thanks for your comments, the suggested changes were made..

Reviewer 2 Report

The structure of the paper fits the scope of the journal, and covers a topic of interest to the journal's audience.

As regards the specific aspects contained in the review, the following points should be noted:
- given the target audience of the journal, the introduction structure addressed effectively the various aspects involving substitution of animal proteins with vegetable sources.
- this reviewer will focus mainly, because of his area of expertise, on sensory aspects. Conceptualization and execution of the CATA experiment is in line with similar approaches for the same materials, as well as choice of multivariate statistical methods chosen. Results give a clear idea of the outcome.
- the conclusions paragraph could benefit from a critical rewriting proposing potential benefits for future research.

Author Response

Response to Reviewer 2 Comments

The structure of the paper fits the scope of the journal, and covers a topic of interest to the journal's audience.

As regards the specific aspects contained in the review, the following points should be noted:
- given the target audience of the journal, the introduction structure addressed effectively the various aspects involving substitution of animal proteins with vegetable sources.
- this reviewer will focus mainly, because of his area of expertise, on sensory aspects. Conceptualization and execution of the CATA experiment is in line with similar approaches for the same materials, as well as choice of multivariate statistical methods chosen. Results give a clear idea of the outcome.
- the conclusions paragraph could benefit from a critical rewriting proposing potential benefits for future research.

Thank you for your feedback, the indicated changes have been made.

Round 2

Reviewer 1 Report

The authors conducted the changes according to reviewer comment.